


# JULES-BE: representation of bioenergy crops and harvesting in the Joint UK Land Environment Simulator vn5.1

Emma W. Littleton[1,2], Anna B. Harper[2,3], Naomi E. Vaughan[4], Rebecca J. Oliver[5], Maria Carolina Duran-Rojas[2,3], Timothy M. Lenton[1,2]

[1] College of Life and Environmental Sciences, University of Exeter, Exeter, EX4 4QE, United Kingdom
[2] Global Systems Institute, University of Exeter, Exeter, EX4 4QE, United Kingdom
[3] College of Engineering, Mathematics and Physical Sciences, University of Exeter, Exeter, EX4 4QE, United Kingdom
[4] Tyndall Centre for Climate Change Research, School of Environmental Sciences, University of East Anglia, Norwich, NR4 7TJ, United Kingdom
[5] Centre for Ecology and Hydrology, Benson Lane, Wallingford, OX10 8BB, United Kingdom

*Correspondence to*: Emma W. Littleton (e.w.littleton@exeter.ac.uk)

**Abstract.** We describe developments to a land surface model, allowing for flexible user-prescribed harvest regimes of various perennial bioenergy crops or natural vegetation types. Our aim is to integrate the most useful aspects of dedicated bioenergy models into dynamic global vegetation models, in order that assessment of bioenergy options can benefit from state-of-the-art Earth system modelling. A new plant functional type (PFT) representing *Miscanthus* is also presented. The *Miscanthus* PFT fits well with growth parameters observed at a site in Lincolnshire, UK; however, global observed yields of *Miscanthus* are far more variable than is captured by the model, suggesting missing model components that influence growth and yields. Global expansion of bioenergy crop areas under a 2 °C emissions scenario and balanced greenhouse gas mitigation strategy from the IMAGE integrated assessment model (RCP2.6-SSP2) achieves a mean yield of 4.3 billion tonnes dry matter per year over 2040–2099, around 30 % higher than the biomass availability projected by IMAGE. In addition to perennial grasses, JULES-BE can also be used to represent short-rotation coppicing; residue harvesting from cropland or forestry; and rotation forestry.

## 1 Introduction

A large supply of biomass energy, from diverse sources, is an essential component of most strategies to avoid dangerous climate change (Rose et al., 2013;Daioglou et al., 2019). Biomass is important both as a versatile energy source (e.g. used for heat and electricity production and transport fuels), and as part of bioenergy with carbon capture and storage (BECCS), the most feasible mechanism by which large amounts of $CO_2$ may be actively removed from the atmosphere (Smith et al., 2015;Bauer et al., 2017;Daioglou et al., 2019).

"Second-generation" bioenergy crops, comprising lignocellulosic perennial grasses, tree species managed as short-rotation coppice, and residues from forestry and agriculture, are the assumed preferred candidates to meet future biomass energy demand (Chum et al., 2011). They are preferred over "first-generation" biofuels such as maize and sugarcane which require





higher nutrient inputs and have undesirable interactions with the food production systems (since they are food crops and must be grown on cropland) (Tilman et al., 2009).

A wide range of estimates of future bioenergy supply exists, but most 2 °C or lower scenarios feature BECCS being rolled out at scale in the next 10–20 years (Fuss et al., 2014;Clarke L. et al., 2014;Rogelj et al., 2018), with bioenergy crops delivering 100–400 EJ year$^{-1}$ (primary energy) by 2100 (Huppmann et al., 2018). The impacts of large-scale bioenergy production on the land surface and Earth system could be significant, because changes to vegetation cover across the Earth can change climate systems through biophysical effects such as changes to albedo, evaporation and runoff, or through biogeochemical effects like disturbance or priming of soil carbon (Fontaine et al., 2004). The importance of bioenergy expansion to future efforts to limit climate change, combined with relative lack of understanding of its environmental effects, strongly motivates further efforts to improve our understanding of this process. Earth system modelling, a method by which we study many aspects of global environmental change, provides a robust framework for simulating and interrogating large-scale land use change such as bioenergy cropland expansion.

Dedicated bioenergy crop models may be used to project yields and responses to environmental stressors at site or regional level (Robertson et al., 2015). MISCANFOR (Hastings et al., 2009) is one example of a *Miscanthus* growth model that has been applied at global scale (Pogson et al., 2013). These models tend to have simple or limited representation of soil carbon cycling, hydrology and climate. Dynamic global vegetation models (DGVMs), by contrast, are models specifically developed to address questions about large-scale vegetation patterns and productivity, and their links with the climate and Earth system (Sitch et al., 2008). However, this typically occurs at the expense of representation of specific plant species and detailed site and management information. There are differences between DGVMs in representation of bioenergy crops and calculation of harvests (Krause et al., 2018): although some DGVMs feature explicit representation of bioenergy crops and harvesting (Beringer et al., 2011;Li et al., 2018b), others use approximations based on generic plant functional types (PFTs) and calculate harvests as a fixed proportion of productivity (Muri, 2018).

Currently the Joint UK Land Environment Simulator (JULES) uses generic C3 and C4 grasses to simulate bioenergy productivity, with harvest taken from 30 % of litter (Harper et al., 2018a). In this paper, we describe new functionality developed within the JULES land surface model to represent the growth and harvest cycles of specific perennial bioenergy crops including lignocellulosic grasses (*Miscanthus*) and trees used in short-rotation coppice regimes (poplar SRC), as well as forest management (Table 1), hereafter called JULES-BE. JULES-BE represents the yield mechanistically by removing the above-ground biomass, reducing the plant's height and leaf area and allowing it to regrow. The parametrisation of a new PFT to represent *Miscanthus* is also presented. The aim of these functional developments is to simulate yields of biomass for energy feedstocks, and to evaluate the impacts of bioenergy cropping on the global carbon cycle and climate system. Therefore, this study fits best with the DGVM approach, which allows analysis of the impacts of bioenergy on climate and land surface processes. JULES has been used to model bioenergy systems before (Hughes et al., 2010;Black et al., 2012;Oliver et al., 2015), at site level, but these approaches have not been integrated into JULES's DGVM, TRIFFID, which links plant productivity to soil carbon and the global carbon cycle. The improved representation of harvesting and yield we present here is unique because





it facilitates the assessment of impacts of bioenergy crops on the carbon cycle and climate system in a way that has not been shown before using the JULES model.

## 2 Technical development

### 2.1 Existing model description

JULES is a community land surface model that can be run standalone (as described here) or used as the land surface component of the Met Office's Earth System models (Collins et al., 2011) . JULES is described in Best et al. (2011) and Clark et al. (2011). JULES calculates the surface energy and water fluxes, along with gross and net primary productivity, on a half-hourly or hourly time step. The net primary productivity (NPP) for each PFT is accumulated during each timestep, to be later used for calculating changes in vegetation structure and coverage in TRIFFID, the dynamic global vegetation model built into JULES.

TRIFFID is called at the end of a user-defined number of days (typically 1 or 10 days), and the accumulated NPP is allocated between "growth" and "spreading." The former is used for increasing leaf area index (LAI) and canopy height, while the latter is used to allow PFTs to take up more space in a grid cell. Competition for space is determined based on PFT heights: the tallest plants get first access to space in a grid cell, but may not be able to compete if their NPP is too low.

In JULES, crops are represented in one of two ways. Major food crops such as wheat, maize and soya are represented by the

JULES-crop module (Osborne et al., 2015). However, JULES-crop is suitable only for annual seed crops, and is not compatible with TRIFFID and the wider carbon cycle representation within JULES. Therefore, the TRIFFID-crop module was developed to represent crops within the carbon cycle and climate system. When the TRIFFID-crop option is enabled within JULES, multiple types of agricultural land are represented separately. The user defines the fraction of each grid cell dedicated to food crops, pasture, and bioenergy. The fractions can vary in time with new values prescribed annually or less frequently. Each of

these crop area types forms a separate "land class" for which specific PFTs are allocated. TRIFFID-crop requires height-based competition (Harper et al., 2018b), which allows for a flexible number of PFTs. Each PFT is assigned to only one land class and competes only with PFTs of the same land class, within the defined fraction. Any land within the fraction that cannot be filled by the assigned PFTs is occupied by bare soil. Multiple identically parametrised PFTs may be used if the same type of plant (e.g., C3 grass) is desired in multiple land classes (e.g., natural, food crop, and pasture). TRIFFID-crop also introduces

harvesting of biomass from crop areas, described in Sec. 2.2.1 as "continuous harvest". JULES-BE describes a set of options within JULES, building upon the TRIFFID-crop functionality to enable periodic harvesting and assisted expansion of bioenergy PFT area.

### 2.2 Harvesting regimes

Two methods of representing crop harvest are used. A new TRIFFID parameter, *harvest_type* (Table 3), may be set to 0, 1 or

2 for each PFT. A value of 0 represents no harvest; the two harvest types are described below.



### 2.2.1 Continuous harvest (type 1)

This harvest type is used and described by Harper et al. (2018a) and represented in Eqs. (1) and (2). A fixed percentage (currently hardcoded as 30 %) of the PFT's litter production ($lit_c$) is rerouted to a harvest pool ($harvest$) on a continuous basis. The remaining litter fraction (currently 70 %) enters the soil pool as normal.

$$harvest = 0.3 \times lit_c \tag{1}$$

$$lit_c = 0.7 \times lit_c \tag{2}$$

### 2.2.2 Periodic harvest (type 2)

At defined intervals, specified in days by the user, the PFT is reduced to a short height, also specified by the user (see Table 2 for a list of parameters). New values for wood ($woodC$), leaf ($leafC$) and root ($rootC$) biomass are calculated based on this height, per Eqs. (46, 56–58, 60) given by Clark et al. (2011) and reproduced in the Supplement. The difference between old and new above-ground carbon is allocated to the harvest pool (Eq. (3)), whereas the change in root (below-ground) carbon is added to the plant litter ($lit_c$), as given in Eq. (4). A time coefficient ($\Delta t$) is used to convert stocks to fluxes.

$$harvest = \frac{(leafC_{t-1} + woodC_{t-1}) - (leafC_t + woodC_t)}{\Delta t} \tag{3}$$

$$lit_c = lit_c + \frac{(rootC_{t-1} - rootC_t)}{\Delta t} \tag{4}$$

Since the model describes a constant perfect correlation between PFT height and balanced-growth LAI, minimum LAI must also be set low enough to accommodate the prescribed *harvest_ht* (Table 2). The PFT then begins to regrow again from its new shorter height.

### 2.3 Assisted expansion

This section describes new functionality which directs the model to simulate planting of new agricultural areas. In the existing scheme, when the fractional area of a land class increases, the new area is covered by bare soil, until the existing vegetation expands into it. Expansion of PFTs in the absence of competition follows Eq. (5). Equation (5) is a simplified version of Eq. (52) in Clark et al. (2011), assuming that only one PFT is assigned to the land class, the PFT occupies at least 1 % of the total grid cell, and the plant has already reached its maximum height. $Cveg$ represents the PFT's biomass density, and $g_{area}$ is a constant parameter representing total mortality.

$$\Delta frac = frac \times \left(\frac{NPP}{Cveg} - g_{area}\right) \tag{5}$$

This arrangement represents competition and growth in natural landscapes, but where land is dedicated to a specific purpose such as bioenergy crops, it is less realistic to represent it as such; it is equivalent to humans clearing an area of land for cropping but then neglecting to plant anything.

Where the agricultural areas consist of ordinary C3 and C4 grasses, this does not pose much of a problem since $Cveg$ is usually small relative to $NPP$ during the growing season; therefore, $\frac{NPP}{Cveg}$ can attain sufficient size to allow the grass to increase its





area. The problem is more significant in the case of high-density lignocellulosic bioenergy grasses, in which $NPP$ may be 1–3 times that of an ordinary grass but $Cveg$ is 5–10 times larger. Annual harvesting also reduces the capacity of crop grasses to increase their area, since more of their $NPP$ is dedicated to increasing their height (i.e., one of the assumptions of Eq. (5) does not hold for much of the year).

Therefore, in order to represent the establishment of new agricultural areas, without sacrificing the benefits of dynamic vegetation, i.e. that plants can die off where the environment is unsuitable, a new planting mechanism has been implemented. This mechanism, activated using the switch *l_ag_expand* globally and the *ag_expand* switch on individual PFTs (Table 2), alters the value of $\Delta frac$ returned by TRIFFID. Land class fractions may change once per year, whereas TRIFFID (where plant competition and fractional allocation takes place) is run once per simulation day. At each grid cell, the current land class

fraction is compared to the value used at the last TRIFFID call. Where the land class fraction has increased, the assisted expansion function is activated. $\Delta frac$ is calculated as it would have been without land use change ($\Delta frac_{na}$ in Eq. (6), which could be positive or negative), but then the value of the increase ($\Delta frac_{ag}$) is added to it. This is equivalent to assuming that agricultural expansion is accompanied by planting new crops. $\Delta frac$ is then added to the previous PFT fraction. If two or more PFTs (for which assisted expansion is enabled) share the same land class, the new area is divided equally between them

($NPFT_{ag}$). This process is also illustrated in Fig. 1.

$$\Delta frac = \Delta frac_{na} + \frac{\Delta frac_{ag}}{NPFT_{ag}} \tag{6}$$

## 2.4 New PFT parametrisation

A new bioenergy PFT was developed representing *Miscanthus*, a perennial grass of particular interest in the bioenergy literature due to its robust growth and low input requirements (Heaton et al., 2008;Zub and Brancourt-Hulmel,

2010;McCalmont et al., 2017). An earlier representation of *Miscanthus* in JULES (Hughes et al., 2010) focused on realistic representation of height and LAI, and estimated yields based on NPP. In the new method of periodic harvesting, above-ground biomass (AGB) is the most important factor determining yields, and therefore this aspect was emphasised in the development of this PFT (Fig. 2(d); Fig S1).

In the current version of JULES, around 90 PFT parameters and 13 TRIFFID parameters govern a PFT's response to its

environment, although they are not all used at once because many parameters are only required by specific configurations. The *Miscanthus* PFT presented here was developed based on a generic C4 grass in the 9 PFT JULES scheme (Harper et al., 2016), with 14 parameters redefined specifically for this study. Table 3 gives an overview of the main features of the *Miscanthus* PFT. A full list of parameters and their relevance in JULES is given in the Supplement.

JULES-BE can represent any type of plant as a bioenergy crop. In addition to perennial grasses, short-rotation coppicing

(SRC) with willow or poplar can be simulated, or softwood or hardwood trees for forestry (Table 1). This study introduces examples of tree types grown for biomass or bioenergy in Sec. 3.4, using two poplar PFTs developed for JULES by Oliver et al. (2015).





## 2.5 Methods of evaluation

### 2.5.1 Lincolnshire site data

Adjustment of PFT parameters for *Miscanthus* was performed using observational data collected from a commercial *Miscanthus* plantation in Lincolnshire, UK. The site is on a compacted loam soil previously used to grow wheat and oilseed rape. The site had mean annual temperature of 9.8 °C and mean annual precipitation of 621 mm. The net ecosystem exchange of $CO_2$ was measured by eddy covariance methodology. Gross primary productivity (GPP) was calculated using the REddyProc method described by Robertson et al. (2017), after Reichstein et al. (2005). Manual measurements of height and LAI were taken over the growing season (Fig. 2).

JULES was driven by meteorological data collected at the site on an hourly basis during 2006–2013 (shortwave and longwave radiation, wind speed, precipitation, temperature, air pressure, and specific humidity). Physical soil properties were derived from measurements taken at the site between 2009 and 2010. The site and data collection are described in greater detail by (Robertson et al., 2016;Robertson et al., 2017).

### 2.5.2 Global bioenergy yield dataset

In order to further explore the suitability of this *Miscanthus* PFT for simulating biomass yields, a comparison was conducted against observed yields. Li et al. (2018a) have compiled a comprehensive global dataset of bioenergy crop yields as reported in scientific literature. It includes 981 observations of *Miscanthus* yields, from the United States and Europe, with and without irrigation and fertiliser.

For comparison with modelled *Miscanthus* yields produced by JULES-BE, the observations of *Miscanthus* from this dataset were combined into 68 0.5°x0.5° grid cells. Observed sites using fertiliser or irrigation were found not to differ significantly in yield from untreated sites, and were therefore included in the comparison. (JULES-BE is not currently configured to support irrigation or nitrogen fertilisation.) JULES-BE was then run at the same 68 grid cells over the period 1980–1999, using meteorological driving data from WATCH at 0.5°x0.5° (Weedon et al., 2010;Weedon et al., 2011).

### 2.5.3 Future simulation

To evaluate implications of the new representation of bioenergy crops for climate mitigation, a 21$^{st}$ century simulation of bioenergy crop area under SSP2-2.6 is shown here. Meteorological driving data from HadGEM2-ES ISIMIP simulations were used, downscaled to 0.5° and bias-corrected to calibrate with WATCH observed climatology over 1960–1999 (Hempel et al., 2013). Atmospheric $CO_2$ concentrations followed the RCP2.6 $CO_2$ concentration pathway, covering the period 2006–2099. The land use scenario is generated by the IMAGE 3.0 integrated assessment model (Stehfest et al., 2014). The RCP2.6-SSP2 scenario (Doelman et al., 2018;Daioglou et al., 2019) features a rapid scale-up of global bioenergy crop area in the tropics over 2025–2045 to around 250 million hectares (Mha), followed by gradual expansion into temperate regions over the rest of the century, with fluctuations in crop area driven by bioenergy demand (Fig. 7). Figure 6, which shows yields across the global





land surface, is generated using the same driving data, though bioenergy crops are not grown on all grid cells in the RCP2.6-SSP2 simulation.

### 2.5.4 Forestry and short-rotation coppice demonstrations

Three simulations were carried out to demonstrate the functionality of JULES-BE for harvesting of woody biomass: short-

rotation coppicing (SRC); permanent (non-felling) forest management with residue harvesting; and rotation forestry plantation. They are presented as illustrative cases to inform future model development, and are thus intentionally idealised scenarios.

These three simulations were carried out for a single point, a FLUXNET site in Italy (IT-CA1, Castel d'Asso; http://sites.fluxdata.org/IT-CA1; Sabbatini et al. (2016)), at which poplar is grown on a short-rotation coppicing regime. Meteorological data was collected onsite from 2011–2014 on a half-hourly basis. Over this period, the mean annual

temperature at this site was 15 °C and the mean annual precipitation was 736 mm. Site soil properties were also used. The local biome (IGBP class) is temperate deciduous forest.

All three simulations were run for a 60-year cycle, using looped meteorological driving data from 2011–14:

- Poplar SRC: two species of Poplar, *Populus nigra* and *P. x euramericana*, parametrised and evaluated by Oliver et al. (2014). Harvesting occurs on a 3-year rotation on day 270 of the year, when trees are cut to 1 metre height. The

PFT and TRIFFID parameters for the poplar PFTs are given in the Supplement.

- Residue harvesting forestry: two tree species; broadleaf deciduous tree and needleleaf evergreen tree. Generic PFT tree parameters as per Harper et al. (2018b) (reproduced in the Supplement). Continuous harvesting (30 % of litter production) is applied to represent residues.

- Rotation forestry: two tree species; broadleaf deciduous tree and needleleaf evergreen tree. Generic PFT tree

parameters as per Harper et al. (2018b) (reproduced in the Supplement), with *lai_min* adjusted to 0.1 to allow for harvest cutting. Harvesting occurs on a 40-year rotation on day 364 of the year, when trees are cut to 1.5 metre height.

## 3 Results

### 3.1 Lincolnshire site

Model results from the Lincolnshire site are shown in Figs. 2(a)–(c) and 3, compared against observational data from the site.

The observations show more year-to-year variation in peak seasonal height and LAI than the model. The modelled peak heights (2.4–2.55 m during 2010–2012) and LAIs (2.75–2.9) are also generally lower than those observed (height: 2.8–3.1 m; LAI: 3.1–4.1), although observed height and LAI tended to decline after their peaks to values closer to those produced by the model. The correlation between observed and modelled GPP at this site is excellent (R=0.956; Fig. 3).

The mean modelled yield was $6.0 \pm 0.5$ tonnes C ha$^{-1}$ year$^{-1}$, equivalent to a dry matter yield of $12.4 \pm 1.1$ tonnes DM ha$^{-1}$

year$^{-1}$ assuming 48 % carbon in dry biomass (Baxter et al., 2014). This significantly exceeds the observed yields of $7.6 \pm 1.6$





tonnes DM ha$^{-1}$ year$^{-1}$ at this site (Robertson et al., 2017), though sits squarely within the range of yields observed in the UK (12.4 ± 5.9 tonnes DM ha$^{-1}$ year$^{-1}$; 11 studies compiled by Li et al. (2018a)).

### 3.2 Modelled *Miscanthus* yields against observations

A comparison of yields was conducted between the JULES-BE model results and observed *Miscanthus* yields compiled from

the literature by Li et al. (2018a). The results of this comparison are given in Figs. 4 and 5.  Across all sites and years, observed yields were much more variable, with a mean ± SD of 12.5 ± 9 tonnes DM ha$^{-1}$ year$^{-1}$ (n=981), compared to 14.3 ± 7 tonnes DM ha$^{-1}$ year$^{-1}$ for the modelled yields (n=1360). In a few cases, yields up to 51 tonnes DM ha$^{-1}$ year$^{-1}$ were observed, exceeding the maximum modelled yield of 37 tonnes DM ha$^{-1}$ year$^{-1}$; but more significantly, low yields of less than four tonnes DM ha$^{-1}$ year$^{-1}$ were much more common in the observations (Fig. 5(b)).

The modelled yields showed a consistent positive correlation with both mean annual precipitation (R=0.752) and mean annual temperature (R=0.718) (Fig. S2). For wider comparison, Fig. 6 shows simulated yields of *Miscanthus* across the global land surface. For the observed yields, the correlation with precipitation was much weaker (R=0.094), and while correlation with mean annual temperature was weak overall (R=0.252), yield appears to peak around 14–15 °C and decline with higher temperatures (Fig. S2).  This difference between modelled and observed results is clearly illustrated in the southern United

States, where modelled yields are as much as 20 tonnes DM ha$^{-1}$ year$^{-1}$ higher than observations (Fig. 4). These observations were of *Miscanthus x giganteus*, a cultivar that produces very high yields in temperate climates but appears less well-adapted to high temperatures (Fedenko et al., 2013). Other perennial grasses may be more appropriate for hot climates. The model PFT would benefit from some further tuning to better represent properties such as stomatal conductance and photosynthetic temperature response photosynthetic temperature response, particularly the *tupp* and *vsl* parameters to better calibrate the

relationship between leaf temperature and maximum rate of carboxylation of Rubisco (*Vcmax*; Sec. S1).
Figure 6 shows modelled yields for the whole Earth area, averaged over 2010–2019, in order to show the general spatial pattern of productivity of *Miscanthus*.  Yields of 8–20 tonnes DM ha$^{-1}$ year$^{-1}$ are typical for most temperate climates, increasing to a maximum of about 35 tonnes DM ha$^{-1}$ year$^{-1}$ in the humid tropics.  Yields are positively correlated with both temperature and precipitation (Fig. S2). This may help to contextualise the yields shown in Fig. 4.

### 3.3 Assisted expansion, global and future yields

To assess the impact of the assisted expansion feature on simulated global *Miscanthus* crop area, Fig. 7 shows total *Miscanthus* crop area in the RCP2.6-SSP2 scenario (van Vuuren et al., 2017). This scenario features a rapid increase in bioenergy crop area ("Available area"; black) from 29 Mha in 2025 to 282 Mha in 2045. "Natural expansion" (green) represents the *Miscanthus* PFT parametrised as discussed here, without using the new agricultural expansion functionality. In this scenario,

*Miscanthus* occupies 13 Mha of the bioenergy crop area in 2025, increasing to 104 Mha in 2045—leaving 178 Mha as bare soil. In 2035, only 31 Mha, or 25 % of the bioenergy crop area, is occupied by *Miscanthus*. With "Assisted expansion" (blue), the *Miscanthus* PFT occupies a consistently larger proportion of the available area throughout this period of rapid increase. In





2035, the PFT covers 119 Mha, 96 % of the available area. The proportion of area covered begins to decline after 2040, as the bioenergy production area shifts from the tropics into temperate biomes which are somewhat less favourable for growth in this representation of *Miscanthus*. The difference in crop area between the old and new expansion methods declines toward the end of the simulation, as the crop area begins to stabilise and the two simulations begin to converge.

In Fig. 8, the total global *Miscanthus* yield is shown, using the "assisted expansion" method shown in Fig. 7. The bioenergy crop yield supplied in the IMAGE model is shown for reference (Huppmann et al., 2018;Doelman et al., 2018;Daioglou et al., 2019).  Following the rapid increase in bioenergy crop area, from 2040-2099, bioenergy crop yields remain fairly steady in JULES-BE at 4.3 Gt DM year$^{-1}$ globally, compared to 3.3 Gt DM year$^{-1}$ in IMAGE over the same period. IMAGE uses a management factor when projecting energy yields, assuming that yields are currently used inefficiently (typical values are 60

% in 2020) but that improvements to crop breeding and management will increase yields to 120–140 % of physical potential by 2100 (Stehfest et al., 2014). This accounts for a portion of the gap in the early years of this scenario which closes between the two models by the 2090s. The *Miscanthus* PFT in JULES-BE probably over-estimates yields in hot climates (Fig. 4); as such, the yields projected by IMAGE may be more reliable. This scenario, and the comparison between JULES-BE and IMAGE, will be explored in greater detail in a future publication.

**3.4 Demonstrations of forestry and short-rotation coppicing**

Figure 9 shows illustrative simulations of short-rotation coppicing and managed forestry using JULES-BE. Over the 20 harvest cycles of poplar SRC, the yield was 2.4 ± 0.3 tonnes C ha$^{-1}$ year$^{-1}$ (*P. Nigra*) and 2.2 ± 0.5 tonnes C ha$^{-1}$ year$^{-1}$ (*P. x Euramericana*).  This falls within the range observed by Sabbatini et al. (2016) over the 2011–2012 growing seasons (3.1 ± 1.5 tonnes C ha$^{-1}$ year$^{-1}$) at the IT-CA1 site (growing *Populus x canadensis* on a 2-year coppicing rotation). The site received

some supplemental irrigation during dry spells, which is not represented in the model; this may account for some under-estimation of yields. For rotation forestry, the yield over the 40-year rotation was 41 t C ha$^{-1}$ for broadleaf and 69 t C ha$^{-1}$ for needleleaf, equivalent to 1.0 and 1.7 t C ha$^{-1}$ year$^{-1}$, respectively. This is higher than the average productivity for European forests (around 0.8 t C ha$^{-1}$ year$^{-1}$, assuming 250 kg C m$^{-3}$ of harvested roundwood) (Payn et al., 2015), but lower than recent estimates from France for Douglas fir of 3.1 t C ha$^{-1}$ year$^{-1}$ following a 40-year rotation (Bréda and Brunette, 2019). These

examples show that with appropriate tuning and validation of the PFT and harvest parameters, JULES-BE could be used to facilitate decision-making on questions such as species selection, harvesting regime, harvest frequency and timing.

**4    Discussion**

The modelled yields of *Miscanthus* were broadly consistent with observations from sites in the USA and Europe, but showed much less variability. A major reason for this is that the harvest frequency is fixed in the model, with no option for irregular

frequency or for harvests to be skipped. For example, in practice *Miscanthus* is generally allowed 1–2 years after planting to establish before being harvested annually, followed by 1–2 years of low yields. The largest yields generally occur during years




4–10 and decline thereafter, with a typical rotation length of 20 years (Zub and Brancourt-Hulmel, 2010). In the model, there is no representation of a plant's age, so it is not possible to establish an age-dependent harvest regime. Another reason for reduced variability in yields is that the root system reverts to the same small size after each harvest (Eq. (4)), dropping its surplus biomass into the soil C pool. In reality, a relatively small proportion of root biomass is shed at harvest, and the mature

plant gets a regrowth benefit from an established root system. The model currently relies on a fixed relationship between above-ground height and root biomass and breaking this link would create other problems in the model relating to PFT scaling. Future versions of JULES will use the Reduced Ecosystem Demography (RED) approach which represents separate mass classes within a PFT (Moore et al., 2018). Alternatively, an approach could be implemented similar to that of Black et al. (2012), in which three PFTs are used to represent different age classes of sugarcane, although this would not be compatible

with dynamic vegetation. Given these difficulties, and the fact that JULES is a global model, accurate average yields with reduced variability compared to observations is likely to be an acceptable compromise for most applications of JULES-BE. The *Miscanthus* PFT has not been tested with other advanced modules within TRIFFID, such as nitrogen cycling or layered soil carbon (Burke et al., 2017), and will likely require additional updating and tuning of parameters to yield useful results with other functions. Since nitrogen content is recorded for the harvested biomass, with appropriate tuning JULES-BE could

also be used to quantify nitrogen loss from bioenergy crop ecosystems due to harvesting.

An example of rotation forestry has been shown in Fig. 9 for a single point. To represent forestry on a country, regional or global scale, further development of the model is required. The harvest frequency and timing are currently fixed for each PFT, meaning that all grid cells are harvested at the same time. Over a large number of grid cells, this would not be realistic and would produce undesirable hydrological and climatic effects. Further improvements to the model could enable the user to

stagger the timing of harvesting. Allowing harvest frequency to vary regionally would better represent rotation forestry and increase yield by enabling the user to choose a regionally appropriate harvest frequency (shorter for more productive regions). Allowing harvest day-of-year to vary regionally would improve global-scale assessment of any bioenergy crop, since harvest timing is dependent on local climatology. This functionality may be best implemented by allowing these variables to be user-prescribed for each grid cell. However, providing these data may be burdensome for the user, and some predictive algorithms

based on climatology and growth, built into the model, may be more appropriate.

The algorithms for competition between PFTs within TRIFFID can potentially be used to determine the most suitable type of bioenergy crop in each grid cell. However, some modifications would need to be made to the existing code. In the simulations presented in this study, TRIFFID competition was enabled, allowing the bioenergy PFT to adjust its area to scale with its productivity—for example, allowing a crop to die back in response to an unsuitable environment. The current competition

scheme is not useful for allowing different types of bioenergy PFTs to compete with each other within a grid cell, since it is based on height. This favours plants that can gain height easily, rather than shorter species with greater biomass density. A competition scheme based on above-ground biomass rather than height would be the first modification to make. This could help select between species within a harvesting regime, e.g. help determine the best perennial grass for annual harvesting, or the best tree species for short-rotation coppicing. However, this may not necessarily select for the highest-yielding plant,





because above-ground biomass is only a good proxy for yield at the end of the growing season, and competition for area is invoked at every iteration of TRIFFID (once per day in these simulations). Also, this development would not be useful for mixing PFTs with different harvest frequency or harvest day-of-year, since it would continue to bias competition towards the PFT that has been harvested less recently. Ultimately, the best solution would be to reapportion the bioenergy crop area

between PFTs once per harvest cycle, based on the previous cycle's yield, but that would be a complex development given the existing model structure.

## 5 Conclusions

This study presents new functionality to represent second-generation bioenergy cropping and harvests in JULES. This is the first step to getting such processes represented mechanistically within Earth system models, in order that the effects of

bioenergy cropping on the carbon cycle and climate system can be evaluated. JULES-BE allows for flexible parametrisation of many types of bioenergy PFTs, although only *Miscanthus* has been fully developed here. Yields of the *Miscanthus* PFT were within the range generally observed in the United States and Europe, though the model failed to capture the large variability in observed yields across and within sites.

Applications for JULES-BE include short-rotation coppicing, rotation forestry and residue harvesting from forests or

agricultural systems. Future development will focus on improving the competition scheme so that multiple bioenergy PFTs can be represented simultaneously, and adding features to the harvest timing mechanism that improve representation of forest harvesting at regional or global scale.

*Code availability.* This work was based on a version of JULES5.1 with additional developments that will be included in a

future release of JULES. The code is available from the JULES FCM repository: https://code.metoffice.gov.uk/trac/jules (registration required). The version used was r12164_biotiles_harvest (located in the repository at branches/dev/emmalittleton/r12164_biotiles_harvest).

*Author contributions.* The aims and objectives of the project were jointly developed by EWL, ABH, NEV and TML. EWL

developed the model code with help from ABH and MCDR and designed and performed the validation simulations with advice from ABH, NEV, RJO and TML. EWL and ABH prepared the manuscript with contributions from NEV, RJO, MCDR and TML.

*Competing interests.* The authors declare that they have no competing interests.


*Acknowledgements.* This work is part of FAB GGR (Feasibility of Afforestation and Biomass energy with carbon capture and storage for Greenhouse Gas Removal), a project funded by the UK Natural Environment Research Council (NE/P019951/1),



part of a wider Greenhouse Gas Removal research programme (http://www.fab-ggr.org/). EWL and ABH also acknowledge support from the UK EPSRC Fellowship EP/N030141/1. MCDR acknowledges support from the CRESCENDO projects that received funding from the European Union's Horizon 2020 research and innovation program under grant number agreement No 641816. Thanks are due to Jon Finch for carrying out the data collection at the Lincolnshire *Miscanthus* site. The authors

thank Eddy Robertson and Andy Wiltshire for providing consultation and advice on the model development. Thanks also to Sarah Chadburn for feedback on draft figures. Maps were produced using map data from naturalearthdata.com.

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





**Table 1. Functionality and applications of JULES-BE.**

| Function | Simulated applications |
|---|---|
| Continuous harvest | Forest management (without felling) |
| | Biomass removal from agricultural land |
| Periodic harvest | Harvests of perennial crops (e.g. *Miscanthus*) |
| | Short rotation coppicing (e.g. poplar) |
| | Forestry rotations |
| Assisted expansion | Planting out of new agricultural areas with bioenergy crops or trees |





**Table 2. TRIFFID parameters required for JULES-BE. An explanation of the use of these parameters is given in Sec. 2.2.**

| Parameter | Type | Values | Definition |
|---|---|---|---|
| *crop* | integer | 0 | Natural land |
| | | 1 | Food crop |
| | | 2 | Pasture |
| | | 3 | Bioenergy crops |
| *harvest_type* | integer | 0 | No harvest |
| | | 1 | Continuous harvest |
| | | 2 | Periodic harvest |
| *harvest_freq* | integer | 0 | Placeholder for harvest types 0 or 1 |
| | | >0 | Interval in days between harvests |
| *harvest_doy* | integer | 0 | Placeholder for harvest types 0 or 1 |
| | | >0 | Day of year on which harvest takes place |
| *harvest_ht* | real | 0 | Placeholder for harvest types 0 or 1 |
| | | >0 | Height to which crop is reduced on harvest [metres] |
| *ag_expand* | integer | 0 | No automatic increase of PFT fraction when land class fraction increases |
| | | 1 | Automatically plant out new crop areas with target PFTs |



**Table 3. PFT parameters distinguishing the *Miscanthus* PFT used in this study. "C4 Grass" parameters are taken from (Harper et al., 2018b); "*Miscanthus* (Hughes)" are given by Hughes et al. (2010). A full list of parameters, with definitions and explanations for values used, is given in the Supplement. "-" indicates parameters that were not yet introduced in the older version of JULES used by Hughes et al. (2010). Parameters described as "Allometry" were determined via an iterative process to improve the relationships**
5 **between above-ground biomass, leaf area index (LAI) and height, as described in the Supplement. Parameters described as "BETYdb" were taken from observations in the Biofuel Ecophysiological Traits and Yields database (LeBauer et al., 2018). "GPP calibration" indicates *tlow* was determined via an iterative process to improve the fit of modelled gross primary productivity (GPP) to the flux data obtained from the Lincolnshire site. "Litter calibration" indicates *g_leaf_0* was determined via an iterative process to approximate the observed ratio of leaf litter to yield (Amougou et al., 2012). Details of these calculations are provided in the**
10 **Supplement.**

| Parameter | C4 Grass (Harper) | *Miscanthus* (Hughes) | *Miscanthus* (this study) | Rationale |
|---|---|---|---|---|
| *a_wl* | 0.005 | 0.014 | 0.07 | Allometry |
| *a_ws* | 1 | 0.9 | 1 | Non-woody plant (100 % live stem) |
| *alpha* | 0.04 | 0.067 | 0.067 | Hughes et al. (2010) |
| *b_wl* | 1.667 | 1.667 | 2 | Allometry |
| *eta_sl* | 0.01 | 0.01 | 0.08 | Allometry |
| *lma* | 0.137 | - | 0.065 | Feng et al. (2012) |
| *tlow* | 13 | 7.85 | 12.8 | GPP calibration |
| *lai_max* | 3 | 3 | 10 | BETYdb |
| *lai_min* | 1 | 0.6 | 0.1 | LAI at minimum height (*harvest_ht* = 0.1) |
| *nmass* | 0.0113 | - | 0.0217 | BETYdb |
| *nr* | 0.0084 | - | 0.0228 | BETYdb |
| *nsw* | 0.0202 | - | 0.0101 | BETYdb |
| *g_leaf_0* | 3 | - | 2 | Litter calibration |



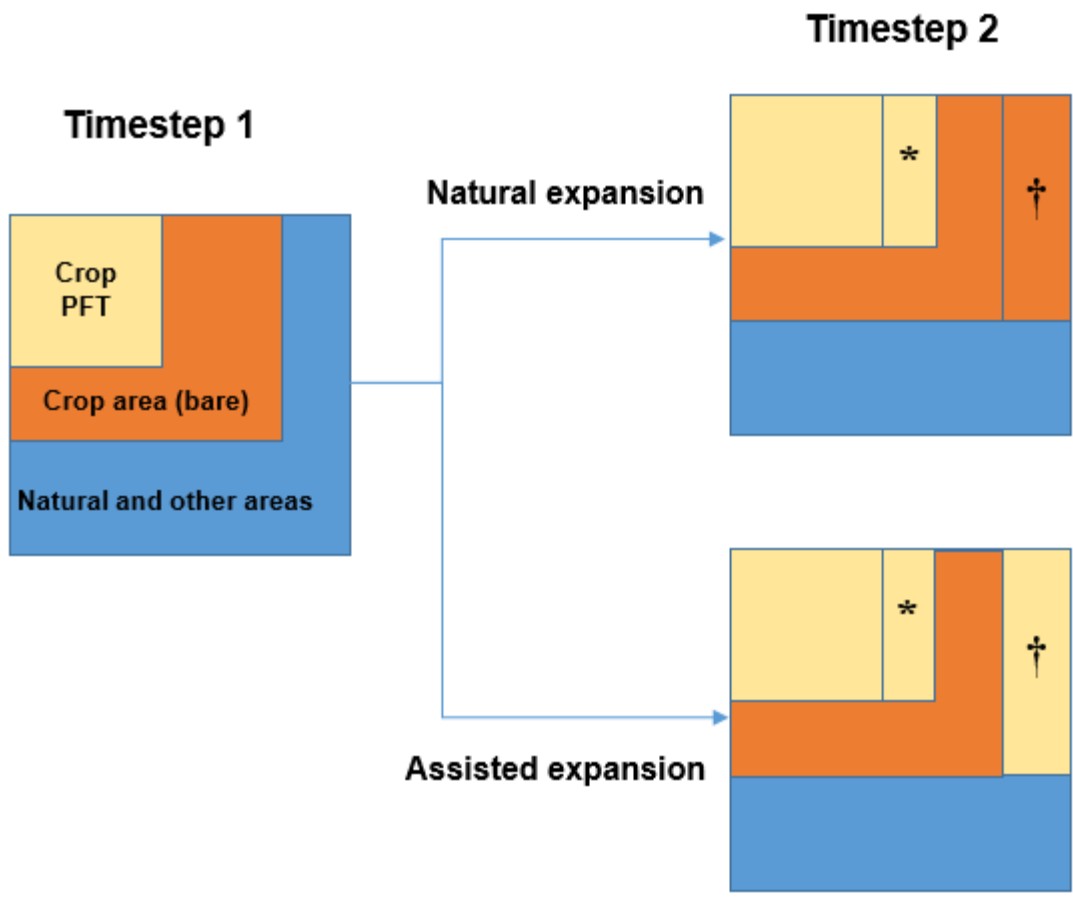

**Figure 1. Schematic of the agricultural expansion functionality in JULES-BE. A full description of the process is provided in Sec. 2.3. The area marked \* represents the change in the plant functional type (PFT) area that would occur without the change in crop area ($\Delta frac_{na}$ in Eq. (6)). The area marked with † represents the newly available agricultural area ($\Delta frac_{ag}$ in Eq. (6)), which is immediately populated with the crop PFT where assisted expansion is enabled, or left bare where it is disabled.**



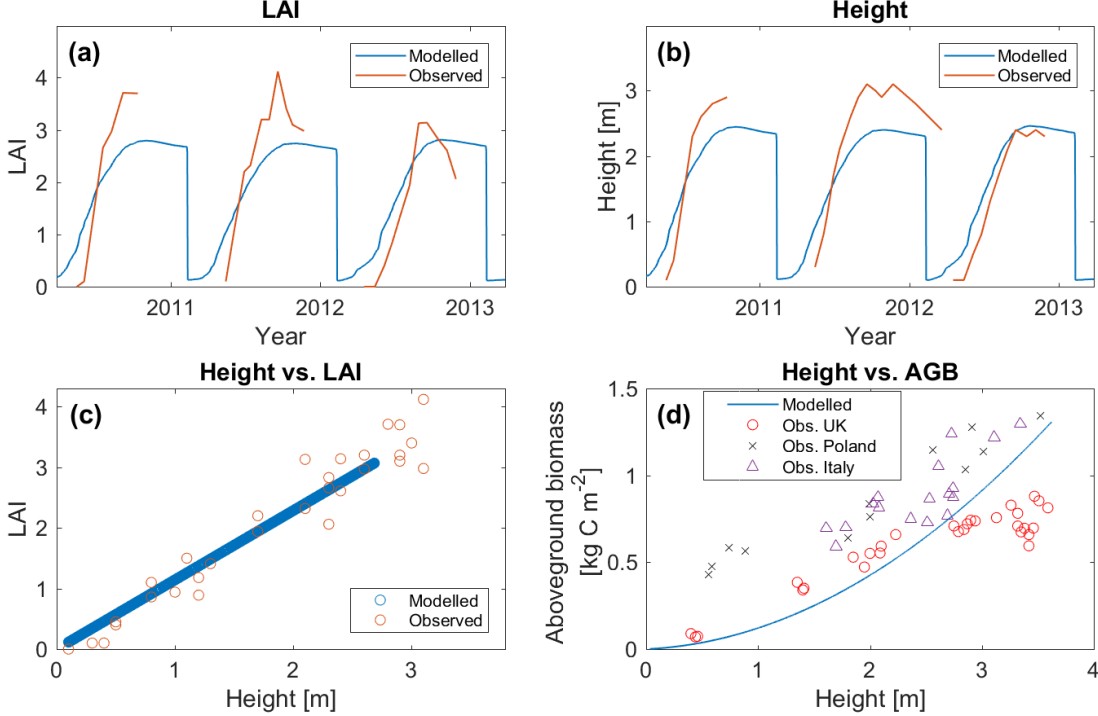

**Figure 2: (a) and (b): Modelled leaf area index (LAI) and height of *Miscanthus*, compared against observations at Lincolnshire, UK, for the period 2010–2013. (c) Relationship between height and LAI, model compared against observations at Lincolnshire, UK. (d) Relationship between height and above-ground biomass (AGB), generic equation from model compared against observations from the UK (Christian et al., 2008), Poland (Jeżowski et al., 2011) and Italy (Cosentino et al., 2007).**



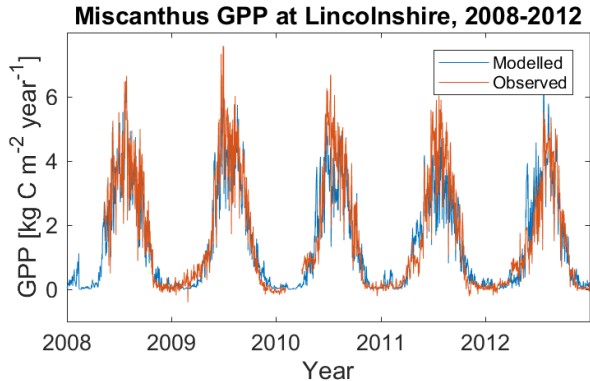

**Figure 3. Modelled gross primary productivity of *Miscanthus*, compared against observations at Lincolnshire, UK, for the period 2008–2012.**



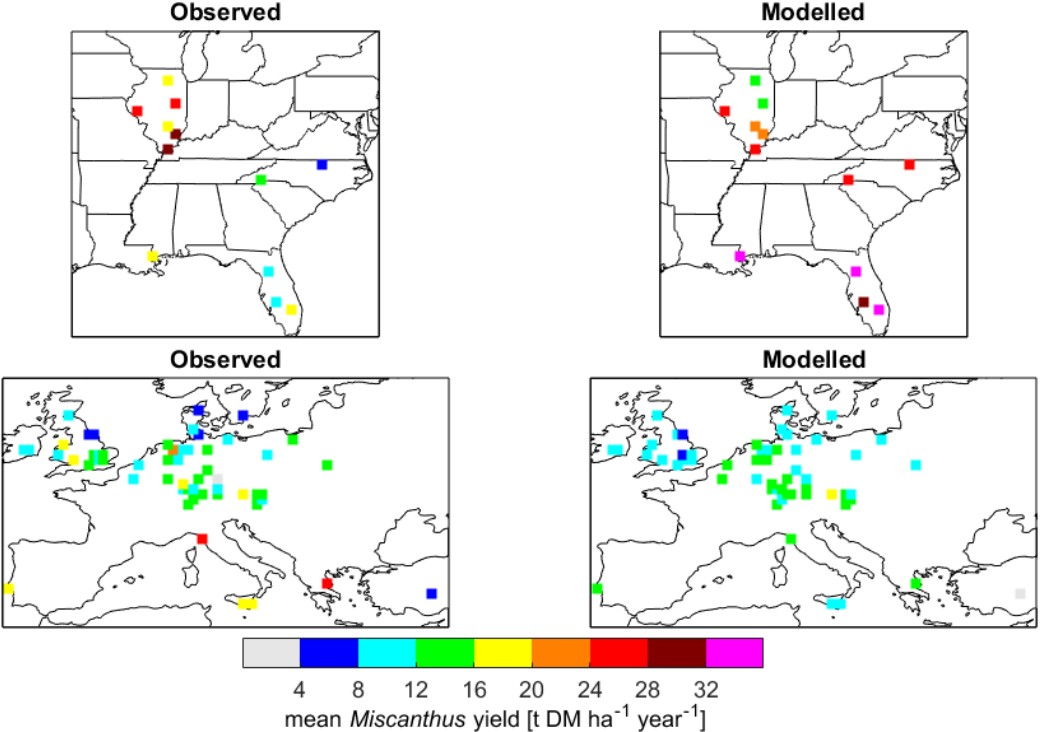

**Figure 4. Comparison of modelled *Miscanthus* yields against observations from Li et al. (2018a).**





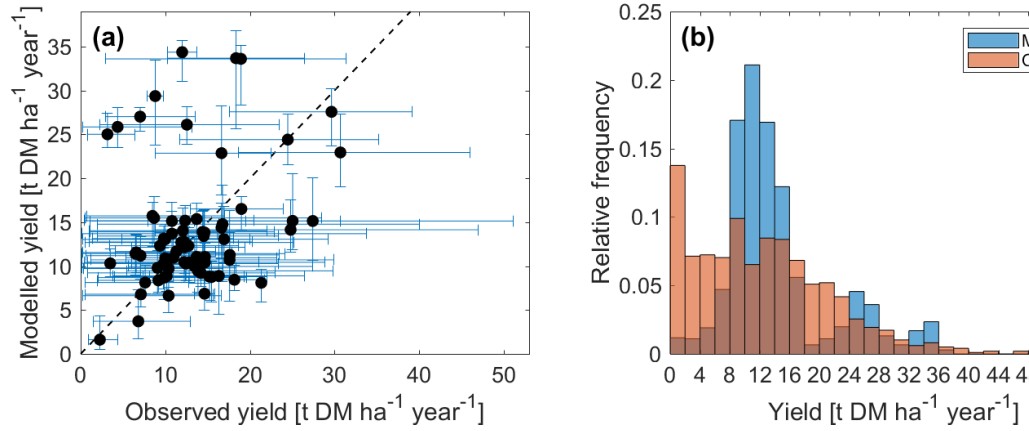

**Figure 5. Modelled yields compared to observed yields of *Miscanthus* collated by Li et al. (2018a). In (a), the error bars give the range of data at each half-degree grid cell. The observed range (horizontal error bars) accounts for variation between sites (different sites may exist within a grid cell area), years, and fertiliser or irrigation treatment; the modelled range (vertical error bars) reflects interannual variability only. (b) shows the range of values by relative frequency. This figure may be compared to Fig. 3(e)–(f) from Li et al. (2018b).**



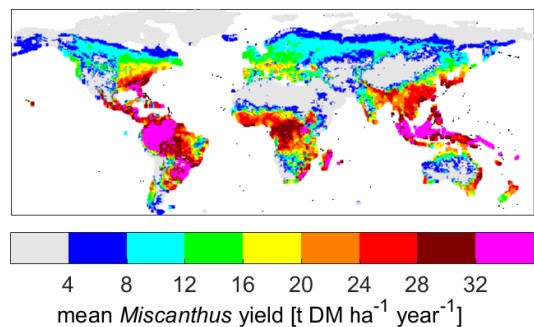

**Figure 6. Modelled yields of the *Miscanthus* PFT, averaged over 2010–2019.**





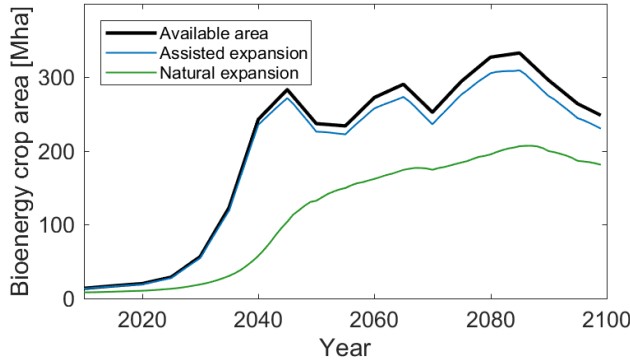

**Figure 7. Modelled area of *Miscanthus* under RCP2.6-SSP2, showing the effect of the agricultural expansion functionality. "Assisted expansion" shows the model run using the assisted expansion function, compared to "Natural expansion" in which this function is disabled. "Available area" shows the total available area for bioenergy crops under this scenario.**



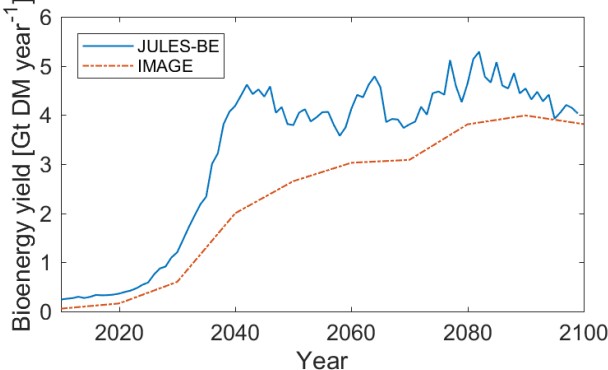

**Figure 8. Bioenergy crop yield from *Miscanthus* under the land-use scenario RCP2.6-SSP2, compared to equivalent bioenergy yield in IMAGE (Huppmann et al., 2018).**



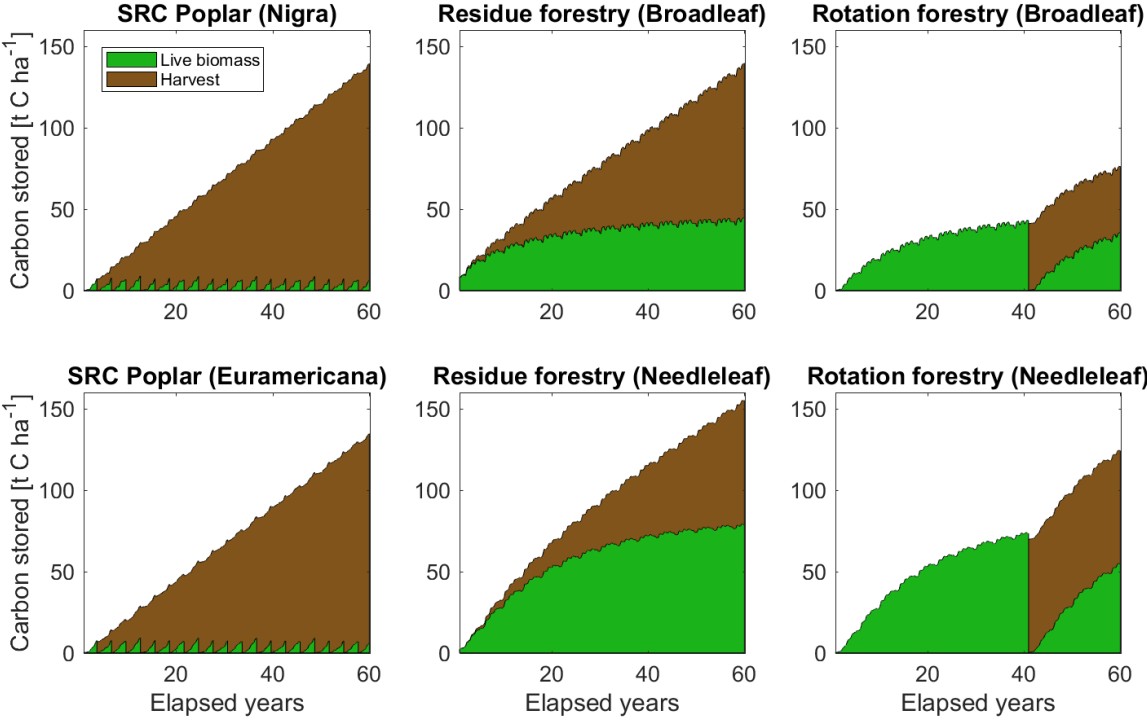

**Figure 9. Illustrated cumulative harvests and vegetation regrowth over a 60-year period, showing the application of JULES-BE to short rotation coppicing (left); harvest of residues from forestry (middle), and rotation forestry (right).**