# Peer review of "JULES-BE: representation of bioenergy crops and harvesting in the Joint UK Land Environment Simulator vn5.1"

_Geoscientific Model Development, 2019_

## Referee Comment (RC1) · Anonymous Referee #1 · 3 Oct 2019

This study implemented Miscanthus into JULES-BE based on site observations from Lincolnshire, UK and a global bioenergy yield dataset. Future simulation was conducted to evaluate the implication of explicit representation of bioenergy crops for climate mitigation. Three simulations were carried to demonstrate the utilization of the new harvesting scheme in JULE-BE.

Miscanthus is one of the most important perennial bioenergy crops that are proposed as biofuel feedstocks for climate mitigation purposes in future scenarios, in particular for the Shared Socio-economic Pathways (SSPs) of the Coupled Model Intercomparison Project Phase 6 (CMIP6). Given the projected biofuel expansion at a global scale,

this study constitutes an important step to explicitly represent bioenergy crops in land surface modeling to assess the implications of large-scale bioenergy expansion on water and carbon cycling.

However, there are a number of suggested comments that should be addressed prior to acceptance. For example, model configurations were not well described. A sensitivity analysis is missing. More discussions on the mismatch between simulation and observation is in need. My main concerns are:

1. Page 2, On line 10-15: Several ESMs have represented bioenergy crops. Therefore, it will be better to conclude and compare what has been implemented in other ESMs.

2. The methods section lacks description for model configuration. For example, it is not clear how the authors chose the initial conditions?

3. Page 5, line 24: Have you done any sensitivity analysis for the parameters?

4. Page 6, line 25: can you justify the consistency between the forcing used to drive the future simulation in this study and the forcing used to drive IMAGE 3.0 to generate the RCP2.6-SSP2 scenario?

5. Page 7, Line 25 and Figure 2, the simulated LAI is much lower than the observations, can you adjust some more parameters based on observations or add more discussions on the potential reasons?

6. Page 6, line 9: Only meteorological and soil properties data were mentioned. It will be great if the authors can also add some descriptions for the validation dataset here. For example, I can see it has some missing periods for the observed LAIs in Figure 2. And how about the land management practice for Miscanthus at this site? Why the durations for soil properties (2009-2010), meteorology (2006-2013), and model validation (2008-2013 for GPP, 2011-2013 for LAI) are inconsistent?

7. Table 3: the authors listed parameters values for C4 grass and Miscanthus for comparison purposes. Can you add more results from C4 grass (e.g., GPP, LAI, NEE)

to have a better sense of the difference between these two PFTs?

8. Figure 2: why the modelled LAI maintained a very high value until the next year and then suddenly became zero (e.g., around Feb 2012)? Especially give GPP did not exhibit similar behaviors in Figure 3.

Specific comments: Page 1, Line 12: it will be better if the authors can specify the model name here rather than at line 20.

Page 1, line 17: what is the missing model component?

Page 2, Line 15: change "global scale" to "global scales"

Page 2, Line 34: what is "TRIFFID"? You only mentioned it later.

Page 6, line 5: it was mentioned that "net ecosystem exchange of $CO_2$ was measured at the Lincolnshire site", so why not show the comparison results for NEE?

Page 6, line 27: it will be great if the authors can specify RCP and SSP.

Page 6, line 10: what are the main soil properties you are concerned with? Do they have significant changes during the simulation period?

Page 9, Line 27: it will be great if you have individual titles for the Discussion part to summarize the main findings and limitations of this study.

Page 11, Line 9: several other studies have implemented bioenergy crops into Earth system models. It should be the first step to get such processes implemented in JULES rather than Earth system models.

Page 11, Conclusion: can you discuss more implications of this study?

Table 2: do you have any ranges for these parameters?

Figure 4: rather than having two subplots show the observation and simulation results, could you add two more figures showing their spatial difference? Or report their spatial correlations?

[Figure]

---

## Referee Comment (RC2) · Anonymous Referee #2 · 3 Oct 2019

General Comments:

The authors outline an enhancing modification of the JULES land surface model, termed JULES-BE, where BE stands for bioenergy. They describe a change to the dynamics of how cropland expands based on the assumption that new cropland will be planted, rather than being filled by the natural expansion of existing cropped area. They show that this change makes the area in bioenergy crops more faithfully conform to that prescribed by the driving IAM scenario in a 21st century simulation. They also present a PFT parameterization for the popular bioenergy crop Miscanthus. This PFT reproduces growth and structural characteristics of Miscanthus for a site in the United

Kingdom but doesn't capture the the full variability of yields observed globally. The PFT also tends to predict unrealistically high yields for hot regions. They also added the ability to simulate coppice, rotation forestry, and litter harvest for bioenergy using existing woody species PFTs. They conclude the paper with demonstration forest bioenergy simulations and initial comparisons to European observations.

The authors convinced me that JULES-BE model represents a useful advancement that will help address important questions in the field. The paper is well written and outlines the technical aspects of the model clearly. I also appreciate the fact that the limitations of model and possible ways to address them are clearly identified. However, there are a few issues in the text that could be clarified or improved, which I detail below.

Specific Comments:

Section 2.2.1 (page 4, line 3):

The rationale of the 30% litter harvest assumption should be briefly described. While this is an existing model assumption the authors chose not to changed it and must therefore feel it is supported. The cited reference does not provide the reasoning for this assumption.

Section 2.5.4 (page 7, line 14):

Explain the rationale for cutting to 1 m in height.

Section 3.1 (page 7-8):

Lines 24-28:

It also seems notable that the model shows onset of growth much earlier than the observations.

Lines 29-30:

The model underestimated the height somewhat for the simulation period (figure 2B) and slightly underestimates the observed aboveground biomass for the UK (Linchonshire) site for the modeled heights (below ∼2.6m, figure 2D). Given this it would seem that aboveground biomass should be low compared to observations. How then does the modeled yield exceed that at the Linconshire site by over 60%?

Section 3.4 (page 9):

This section is underdeveloped. While the authors do make it clear that the simulations are mainly proof of principle they will be of considerable interest to many readers as they demonstrate the culmination of the model changes presented. In particular, the residue forestry panels in figure 9 are not even mentioned. These results suggest that litter harvest can provide roughly the same biomass yield as coppice while having very little impact on forest growth (comparing to the first 40 years of the rotation panels). This is a very provocative initial result and should be contextualized in the text as is done for the coppice and rotation simulations.

Section 4 (page 10):

Sentence line 22-23:

The interpretation of this sentence depends on the definition of 'crop'. Throughout the paper the term crop is used generically with section 2.4 explicitly stating "JULES-BE can represent any type of plant as a bioenergy crop" and in a few places is explicitly qualified, e.g. 'crop grasses'. Please clarify the meaning here. If the statement pertains only to annual crops like grasses I accept the conclusion. However, if trees are included in the definition of crops I would expect that the day of harvest has some potential to impact yield of short rotation coppice but will have very limited impact on predicted yield for longer forest rotations.

Last paragraph starting line 26:

I am not convinced by the authors' contention that the TRIFFID completion scheme

can be made to inform the choice of bioenergy crops appropriate to a given location. The authors present potential changes to the competition scheme that, if I am reading it correctly, would allow PFTs placed in the same land class to compete on the basis of aboveground biomass and / or post season yield calculations. Even if these changes were made it is not clear how this would add greater insight than performing independent simulations with potential PFTS and comparing yields directly. More fundamentally yields do not seem to be the appropriate metic for comparing bioenergy crops in the context of an ESM. If yields were the main concern species specific crop models would probably be sufficient for this purpose. While yield is certainly important for the economics of species selection, it is not sufficient for climate relevance. The value of an ESM is that it allows the impact of bioenergy crops to be examined holistically. Assessing alternatives requires considering the status of carbon stocks and biophysical feedbacks alongside the offset of emissions from crop yields. I do think JULES-BE will be useful in performing such an analysis, just not in the manner described here.

Figures 2 and S1:

Consider providing goodness of fit statistics for figure 2C, 2D, and for at least the selected model (case 1) in figure S1.

Technical Corrections:

Section 2.2.2. (page 4, line 4):

For consistency with the remainder of the formula litC, on both sides of the equation, should have time subscripts.

―――――――――――――――

---

## Author Comment (AC1) · 21 Nov 2019

**Review by Anonymous Referee #1 & author's response**

This study implemented Miscanthus into JULES-BE based on site observations from Lincolnshire, UK and a global bioenergy yield dataset. Future simulation was conducted to evaluate the implication of explicit representation of bioenergy crops for climate mitigation. Three simulations were carried to demonstrate the utilization of the new harvesting scheme in JULE-BE.

Miscanthus is one of the most important perennial bioenergy crops that are proposed as biofuel feedstocks for climate mitigation purposes in future scenarios, in particular for the Shared Socio-economic Pathways (SSPs) of the Coupled Model Intercomparison Project Phase 6 (CMIP6). Given the projected biofuel expansion at a global scale, this study constitutes an important step to explicitly represent bioenergy crops in land surface modeling to assess the implications of large-scale bioenergy expansion on water and carbon cycling.

However, there are a number of suggested comments that should be addressed prior to acceptance. For example, model configurations were not well described. A sensitivity analysis is missing. More discussions on the mismatch between simulation and observation is in need.

My main concerns are:
Page 2, on line 10-15: Several ESMs have represented bioenergy crops. Therefore, it will be better to conclude and compare what has been implemented in other ESMs.

> This section discusses BE crops in ESMs under the general banner of "DGVMs". Adjusted the language in this paragraph to make this more specific. This now reads:
>
> "Dynamic global vegetation models (DGVMs), by contrast, are models specifically developed to address questions about large-scale vegetation patterns and productivity, and their links with the climate and Earth system (particularly as part of the Earth system models of which they form the terrestrial components) (Sitch et al., 2008). However, this typically occurs at the expense of representation of specific plant species and detailed site and management information. There are differences between DGVMs (and ESMs) in representation of bioenergy crops and calculation of harvests (Krause et al., 2018): although some feature explicit representation of bioenergy crops and harvesting (e.g. LPJml (Beringer et al., 2011;Boysen et al., 2016); ORCHIDEE-MICT-BIOENERGY (Li et al., 2018b)), others use approximations based on generic plant functional types (PFTs) and calculate harvests as a fixed proportion of productivity (e.g. NorESM (Muri, 2018))."

The methods section lacks description for model configuration. For example, it is not clear how the authors chose the initial conditions?

> Added this explanatory paragraph at the start of Section 2.5:
> "Simulations were carried out to evaluate and illustrate the new functionality in JULES-BE. These simulations were all based on the JULES-ES configuration, a set of options designed for best representation of carbon cycle and climate dynamics over decadal to centennial timescales (ref). All simulations began with initial conditions from a spin-up to equilibrium, then included a transient spin-up period prior to the main run."

Page 5, line 24: Have you done any sensitivity analysis for the parameters?

PFT parameter choice is discussed in the Supplement and in Harper et al. (2018b).

Added this sentence to Section 2.4:
"(See also Harper et al. (2018b) for further information about PFT parameter selection.)"

Page 6, line 25: can you justify the consistency between the forcing used to drive the future simulation in this study and the forcing used to drive IMAGE 3.0 to generate the RCP2.6-SSP2 scenario?

The future simulation is driven by modelled meteorology from HadGEM2-ES, forced by the RCP2.6 concentrations of CO2 and other radiative forcing agents, and is therefore consistent with the IMAGE scenario of RCP2.6.

Added "(for RCP2.6)" to Page 6, line 26, to improve clarity.

Page 7, Line 25 and Figure 2, the simulated LAI is much lower than the observations, can you adjust some more parameters based on observations or add more discussions on the potential reasons?

Section 2.4 discusses PFT parametrisation, with further details provided in the Supplement. Leaf mass per area and leaf nitrogen concentration were taken from literature. Values for variables governing *vcmax* were determined via iteration to fit the GPP observations at this site. PFT allometry was optimised for height to above-ground biomass relationship.

Page 6, line 9: Only meteorological and soil properties data were mentioned. It will be great if the authors can also add some descriptions for the validation dataset here. For example, I can see it has some missing periods for the observed LAIs in Figure 1. And how about the land management practice for Miscanthus at this site? Why the durations for soil properties (2009-2010), meteorology (2006-2013), and model validation (2008-2013 for GPP, 2011-2013 for LAI) are inconsistent?

Added further detail to Section 2.5.1 for increased clarity. These paragraphs now read:

"Adjustment of PFT parameters for Miscanthus was performed using observational data collected from a commercial Miscanthus plantation in Lincolnshire, UK. The site is on a compacted loam soil previously used to grow wheat and oilseed rape. The site had mean annual temperature of 9.8 °C and mean annual precipitation of 621 mm.  The net ecosystem exchange of CO2 was measured by eddy covariance methodology. Gross primary productivity (GPP) was calculated using the REddyProc method described by Robertson et al. (2017), after Reichstein et al. (2005).  Manual measurements of height and LAI were taken over the growing season (Fig. 2). The site practices, harvesting regime and data collection are described by Robertson et al. (2016) and Robertson et al. (2017).

"JULES requires meteorological and soil ancillary (time-invariant) data to drive the model. Meteorological data were collected at the site on an hourly basis during 2006–2013 (shortwave and longwave radiation, wind speed, precipitation, temperature, air pressure, and specific humidity). Physical soil properties (soil albedo, heat capacity, thermal

conductivity, hydraulic conductivity at saturation, soil moisture at saturation, soil moisture at critical point, soil moisture at wilting point, Brooks-Corey exponent for soil hydraulic calculations, soil matric suction at saturation) were derived from measurements taken at the site between 2009 and 2010."

Table 3: the authors listed parameters values for C4 grass and Miscanthus for comparison purposes. Can you add more results from C4 grass (e.g., GPP, LAI, NEE) to have a better sense of the difference between these two PFTs?

> Added a Figure S3 to the Supplement showing GPP, NPP, height and LAI at the Lincolnshire site for Miscanthus and C4 grass.

Figure 2: why the modelled LAI maintained a very high value until the next year and then suddenly became zero (e.g., around Feb 2012)? Especially give GPP did not exhibit similar behaviors in Figure 3.

> LAI is suddenly reduced to near zero in February at the point of harvesting, because LAI scales with height (except where deciduous behaviour is present, but this is not the case for the Miscanthus PFT).

> This explanation has been added to Section 3.1:

> "The seasonal cycle of growth through to harvest in mid-February is illustrated by the seasonal fluctuation of height and LAI (Figs. 2(a)-(b))."

Specific comments:
Page 1, Line 12: it will be better if the authors can specify the model name here rather than at line 20.

> The sentence now begins "We describe developments to the land surface model JULES …"

Page 1, line 17: what is the missing model component?

> Changed the text on this line from:

> "…suggesting missing model components that influence growth and yields"

> to:

> "…primarily owing to the model's lack of representation of crop age and establishment time."

Page 2, Line 15: change "global scale" to "global scales"

> Changed to "…a global scale" as I believe this is more accurate.

Page 2, Line 34: what is "TRIFFID"? You only mentioned it later.

> TRIFFID was defined in this sentence, which has been rearranged for greater clarity, and now reads:

> "…but these approaches have not been integrated into TRIFFID, the DGVM within JULES which links plant productivity to soil carbon and the global carbon cycle."

Page 6, line 5: it was mentioned that "net ecosystem exchange of CO2 was measured at the Lincolnshire site", so why not show the comparison results for NEE?

> This is possible to do. However, NEE includes soil carbon respiration, which is heavily dependent on soil carbon content. The model was not specifically initialised with observed soil carbon density since soil carbon is a diagnostic variable in the model. Therefore while a comparison between observed and modelled NEE is possible, uncertainty and inconsistencies between modelled and observed soil respiration rates would prevent this analysis from providing clarity about the Miscanthus PFT.

Page 6, line 27: it will be great if the authors can specify RCP and SSP.

> Reference to the $CO_2$ concentration corresponding to RCP2.6 now specifies RCP2.6-SSP2 derived from IMAGE. There are minor differences in $CO_2$ concentration between different SSPs and models within an RCP (in the SSP database https://tntcat.iiasa.ac.at/SspDb).

Page 6, line 10: what are the main soil properties you are concerned with? Do they have significant changes during the simulation period?

> Soil properties are prescribed at the start of the model run and do not vary in time. They are used in the calculations of soil hydrology and thermal conductance.

> Added the list of soil ancillaries in a parenthesis in Section 2.5.1.

Page 9, Line 27: it will be great if you have individual titles for the Discussion part to summarize the main findings and limitations of this study.

> Separated Discussion into "Main findings and limitations" and "Further work".

Page 11, Line 9: several other studies have implemented bioenergy crops into Earth system models. It should be the first step to get such processes implemented in JULES rather than Earth system models.

> I stand by the wording used: "This is the first step to getting **such processes** represented **mechanistically** within Earth system models."
> The emphasis here is on having the relevant physical properties (such as the crop's height at various times of the year, carbon harvested from the system at appropriate intervals) represented so we can explore the effects on the carbon cycle and climate system.

Page 11, Conclusion: can you discuss more implications of this study?

Added the following paragraph at the end of Section 5:
"Implications of this model functionality include the ability to study bioenergy cropping and harvests within a land surface model. Ultimately, this should facilitate climate change mitigation and climate modelling research to evaluate future low-carbon energy systems featuring bioenergy crops for their impacts on hydrology, climate and carbon storage."

Table 2: do you have any ranges for these parameters?

[I assume the reviewer is referring to Table 3, since Table 2 shows TRIFFID parameters and lists allowed values.]
PFT parameter choice is discussed in the Supplement and in Harper et al. (2018b).

Added this sentence to Section 2.4:
"(See also Harper et al. (2018b) for further information about PFT parameter selection.)"

Figure 4: rather than having two subplots show the observation and simulation results, could you add two more figures showing their spatial difference? Or report their spatial correlations?

The difference between modelled and observed yields has been added to Figure 4.

---

## Author Comment (AC2) · 21 Nov 2019

**Review by Anonymous Referee #2 & author's response**

General Comments:

The authors outline an enhancing modification of the JULES land surface model, termed JULES-BE, where BE stands for bioenergy. They describe a change to the dynamics of how cropland expands based on the assumption that new cropland will be planted, rather than being filled by the natural expansion of existing cropped area. They show that this change makes the area in bioenergy crops more faithfully conform to that prescribed by the driving IAM scenario in a 21st century simulation. They also present a PFT parameterization for the popular bioenergy crop Miscanthus. This PFT reproduces growth and structural characteristics of Miscanthus for a site in the United Kingdom but doesn't capture the the full variability of yields observed globally. The PFT also tends to predict unrealistically high yields for hot regions. They also added the ability to simulate coppice, rotation forestry, and litter harvest for bioenergy using existing woody species PFTs. They conclude the paper with demonstration forest bioenergy simulations and initial comparisons to European observations.

The authors convinced me that JULES-BE model represents a useful advancement that will help address important questions in the field. The paper is well written and outlines the technical aspects of the model clearly. I also appreciate the fact that the limitations of model and possible ways to address them are clearly identified. However, there are a few issues in the text that could be clarified or improved, which I detail below.

Specific Comments:

Section 2.2.1 (page 4, line 3):

The rationale of the 30% litter harvest assumption should be briefly described. While this is an existing model assumption the authors chose not to changed it and must therefore feel it is supported. The cited reference does not provide the reasoning for this assumption.

> Added the following explanation to Section 2.2.1:
> "Setting the harvest rate to 30 % of litter production approximates the estimate of 8.2 Pg C year$^{-1}$ of human-appropriated net primary production from crop harvests globally in 2000 (Haberl et al., 2007). Future development of JULES-BE will allow the harvest rate to be user-prescribed for each PFT."

Section 2.5.4 (page 7, line 14):

Explain the rationale for cutting to 1 m in height.

> 1 metre height was used as an illustrative example. In short-rotation coppicing, thin stems are cut near the base from a thicker trunk, rather than to a specific height. 1 metre height equates to an above-ground biomass of 117 g C m$^{-2}$ for *P. nigra* and 153 g C m$^{-2}$ for *P. x euramericana*. Shorter cutting height would result in longer re-establishment times.

> The relevant sentence in section 2.5.4 has been updated and now reads:
> "Harvesting occurs on a 3-year rotation on day 270 of the year, when trees are cut to 1 metre height, allowing sufficient remaining biomass for rapid regrowth the following year."

Section 3.1 (page 7-8):

Lines 24-28: It also seems notable that the model shows onset of growth much earlier than the observations.

>Agreed – this is worth mentioning.

>Added this sentence to Section 3.1:
>"The modelled crop also increased in height and LAI earlier in the season compared to observations."

Lines 29-30: The model underestimated the height somewhat for the simulation period (figure 2B) and slightly underestimates the observed aboveground biomass for the UK (Linchonshire) site for the modeled heights (below _2.6m, figure 2D). Given this it would seem that aboveground biomass should be low compared to observations. How then does the modeled yield exceed that at the Linconshire site by over 60%?

>Added the following text to the end of Section 3.1:

>"The model underestimated height during the growing season but overestimated the yields. This suggests that ratio of height to aboveground biomass was lower at this site than the sites used for calibration in Figs. 2(d) and S1. However, height at harvest time was not recorded; peak height occurred around August to September while harvest was in February or early April. It is usual for Miscanthus to lose biomass over autumn and winter; the preference for harvesting in mid/late winter is not for largest yields but for improved fuel quality and reduced nitrogen loss from the system."

Section 3.4 (page 9):
This section is underdeveloped. While the authors do make it clear that the simulations are mainly proof of principle they will be of considerable interest to many readers as they demonstrate the culmination of the model changes presented. In particular, the residue forestry panels in figure 9 are not even mentioned. These results suggest that litter harvest can provide roughly the same biomass yield as coppice while having very little impact on forest growth (comparing to the first 40 years of the rotation panels). This is a very provocative initial result and should be contextualized in the text as is done for the coppice and rotation simulations.

>We will find an appropriate case in the literature to inform a new simulation of residue forestry to strengthen this proof of principle.

Section 4 (page 10):
Sentence line 22-23:
The interpretation of this sentence depends on the definition of 'crop'. Throughout the paper the term crop is used generically with section 2.4 explicitly stating "JULES-BE can represent any type of plant as a bioenergy crop" and in a few places is explicitly qualified, e.g. 'crop grasses'. Please clarify the meaning here. If the statement pertains only to annual crops like grasses I accept the conclusion. However, if trees are included in the definition of crops I would expect that the day of harvest has some potential to impact yield of short rotation coppice but will have very limited impact on predicted yield for longer forest rotations.

I agree that for forestry, altering the harvest day-of-year would have little impact on yield, but would affect other ecosystem properties. This sentence has been updated and now reads:

"Allowing harvest day-of-year to vary regionally would improve global-scale assessment of any bioenergy crop, as harvest timing is dependent on local climatology and affects local land-surface properties, such as roughness length, albedo, and transpiration rate, which in turn affect the climate."

Last paragraph starting line 26:
I am not convinced by the authors' contention that the TRIFFID completion scheme can be made to inform the choice of bioenergy crops appropriate to a given location. The authors present potential changes to the competition scheme that, if I am reading it correctly, would allow PFTs placed in the same land class to compete on the basis of aboveground biomass and / or post season yield calculations. Even if these changes were made it is not clear how this would add greater insight than performing independent simulations with potential PFTS and comparing yields directly. More fundamentally yields do not seem to be the appropriate metric for comparing bioenergy crops in the context of an ESM. If yields were the main concern species specific crop models would probably be sufficient for this purpose. While yield is certainly important for the economics of species selection, it is not sufficient for climate relevance. The value of an ESM is that it allows the impact of bioenergy crops to be examined holistically. Assessing alternatives requires considering the status of carbon stocks and biophysical feedbacks alongside the offset of emissions from crop yields. I do think JULES-BE will be useful in performing such an analysis, just not in the manner described here.

Thanks for this interesting and thoughtful critique. I do agree that the DGVM competition mechanism may never be an appropriate instrument for evaluating suitability or preferability of different BE crops in the same grid cell or bioregion.

Added the following sentence to the end of Section 4:
"Ultimately however, a yield-based competition scheme would still ignore the biophysical, economic and environmental factors that influence choice of crop type. As such, JULES-BE may always be more useful for informing these land-use decisions based on its output, rather than integrating these decisions into the existing model."

Figures 2 and S1:
Consider providing goodness of fit statistics for figure 2C, 2D, and for at least the selected model (case 1) in figure S1.

The root mean square errors of the modelled relationships to observations have been added for Case 1 of Fig S1 (all panels). These are the same relationships as in Figs. 2(c)-(d), so they have not been added to Figure 2.

Technical Corrections:
Section 2.2.2. (page 4, line 4): For consistency with the remainder of the formula litC, on both sides of the equation, should have time subscripts.

Added time subscripts to $lit_c$ and *harvest* in Eqs. (1)-(4).